# Catalytic undirected methylation of unactivated C(sp³)−H bonds suitable for complex molecules

Jin-Fay Tan [1], Yi Cheng Kang [1] & John F. Hartwig [1] ✉

In pharmaceutical discovery, the "magic methyl" effect describes a substantial improvement in the pharmacological properties of a drug candidate with the incorporation of methyl groups. Therefore, to expedite the synthesis of methylated drug analogs, late-stage, undirected methylations of C(sp³)−H bonds in complex molecules would be valuable. However, current methods for site-selective methylations are limited to activated C(sp³)−H bonds. Here we describe a site-selective, undirected methylation of unactivated C(sp³)−H bonds, enabled by photochemically activated peroxides and a nickel(II) complex whose turnover is enhanced by an ancillary ligand. The methodology displays compatibility with a wide range of functional groups and a high selectivity for tertiary C−H bonds, making it suitable for the late-stage methylation of complex organic compounds that contain multiple alkyl C−H bonds, such as terpene natural products, peptides, and active pharmaceutical ingredients. Overall, this method provides a synthetic tool to explore the "magic methyl" effect in drug discovery.

The "magic methyl" effect is a phenomenon in medicinal chemistry wherein the incorporation of a simple methyl group in a biologically active molecule leads to a substantial increase in pharmacological activity[1–6]. This outcome is attributed to the alterations of several physicochemical parameters, including, but not limited to, metabolic stability[7], binding affinities[8], conformational flexibility[9], and energetics of desolvation[10]. One notable example is Simvastatin, a well-known cholesterol-lowering drug that has a greater potency than that of Mevastatin and Lovastatin (Fig. 1a, left)[11]. Such an effect has led, in numerous pharmaceutical ingredients, to a 100−1000-fold increase in potency after installation of a methyl group (Fig. 1a, right)[1,2]. Consequently, methylated analogs of biologically active compounds have become targets in drug discovery campaigns. However, the preparations of these methylated analogs often require laborious and time-intensive de novo syntheses. As a result, late-stage methylations of C−H bonds at sp³ and sp² carbon centers, particularly those without the need for a directing group[12–15], would be valuable.

While methylations of C(sp²)−H bonds have been extensively investigated[12,14], the undirected methylation of C(sp³)−H bonds is a much more challenging reaction to achieve. Various researchers have reported the methylation of activated C(sp³)−H bonds located α to a heteroatom or at a benzylic position (Fig. 1b)[16–23]. While these advancements are notable, the site-selective methylation of unactivated C(sp³)−H bonds remains an unsolved synthetic problem. One relevant report is Li's methylations of hydrocarbons (Scheme 1c)[24]. However, this transformation is not selective for methylations of primary, secondary, and tertiary C(sp³)−H bonds. Therefore, it is not applicable to the site-selective methylation of structurally complex biologically active compounds. Recently, Stahl reported the undirected methylation of C−H bonds located α to a heteroatom or at a benzylic position with alkyl peroxides and a nickel catalyst under photolytic conditions[17]. While this protocol leads to the methylation of activated C(sp³)−H bonds, it is not generally applicable for the methylation of unactivated C(sp³)−H bonds (vide infra).

Building on our group's Cu- and Ni-catalyzed oxidation and amination of unactivated C−H bonds with peroxide reagents[25–27], as well as our catalytic azidation and halogenation of tertiary C−H bonds of complex molecules with Zhdankin's λ³-azidoiodane[28–30], we report a nickel-catalyzed site-selective methylation of unactivated tertiary C(sp³)−H bonds with peroxides activated photochemically that is

[1]Department of Chemistry, University of California, Berkeley, CA, USA. ✉e-mail: jhartwig@berkeley.edu

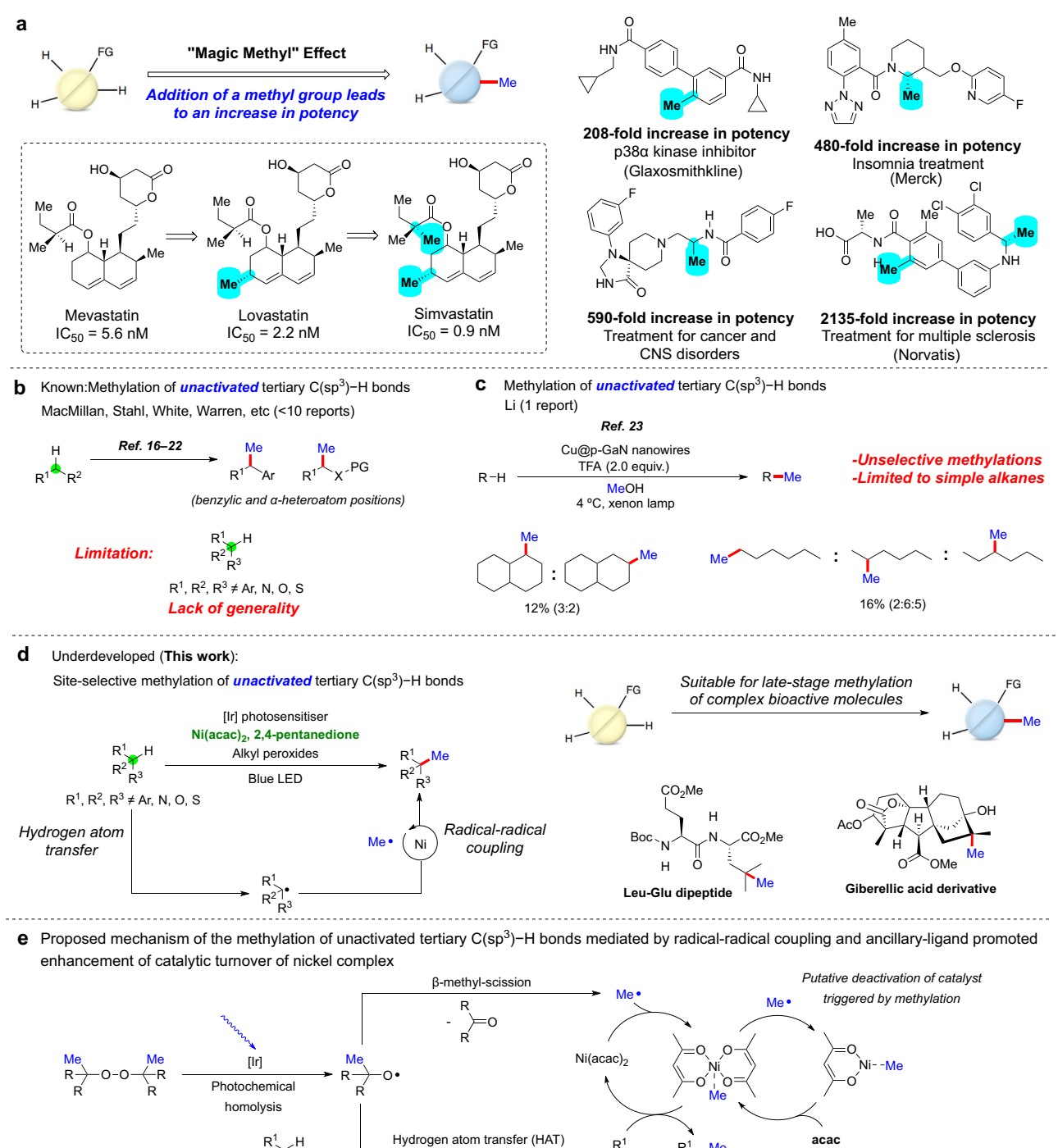

**Fig. 1 | The "magic methyl" effect in medicinal chemistry and undirected methylations of C(sp³)−H bonds. a** Selected examples of drug molecules with an increase in biological activity after installation of methyl groups in the highlighted positions. The methyl groups are highlighted in cyan. **b** The undirected methylation of activated C(sp³)−H bonds is reported, but the site-selective methylation of unactivated C(sp³)−H bonds is not known. The sites of C−H functionalization is highlighted in green; the installed methyl group is shown in blue. **c** The drawbacks of Li's methylation include functionalizations that are unselective and a substrate scope that is limited to alkanes. **d** The site-selective undirected methylation of unactivated tertiary C(sp³)−H bonds presented in this work is suitable for the late-stage methylation of structurally complex organic molecules. **e** Proposed mechanism of Ni-catalyzed formation of C(sp³)−methyl bond mediated by photosensitization and ligand-promoted increase in the turnover of the nickel catalyst.

applicable to the methylation of complex organic molecules and natural products (Fig. 1d). The success of this transformation rests on three design features: (1) a sequence comprising the generation of alkoxy radicals from the photolysis of alkyl peroxides under mild conditions, abstraction of a hydrogen atom from a tertiary C(sp³)−H

bond, and parallel generation of Me• by β-methyl-scission; (2) the formation of a tertiary C(sp³)−methyl bond catalyzed by a nickel(II) complex; and (3) an increase in the turnover of the nickel catalyst by an appropriate ancillary ligand (Fig. 1e). The reaction is compatible with a wide array of functional groups, making it suitable for the late-stage

**Table 1 | Initial studies on the undirected methylation of tertiary C(sp³)−H bonds catalyzed by transition metals**

Ir-F (1 mol%)
Catalyst (5 mol%), ligand (5 mol%)
Dicumyl peroxide (6 equiv.)
MeCN, Blue LED
25 °C, 16 h

1a → 2a

| Entries | Catalyst | Ligand | Additive (0.5 equiv.) | Conversion (%) | Yield (%) |
|---|---|---|---|---|---|
| 1 | NiCl₂.diglyme | L1, L2, L3, or L4 | - | 30–40 | <5 |
| 2[a] | NiCl₂.diglyme | L1 | TFA | 18 | <5 |
| 3 | NiCl₂.diglyme | L1 | B(OH)₃ | 29 | <5 |
| 4 | NiCl₂.diglyme | L3 | MeB(OH)₂ | 31 | <5 |
| 5 | Ni(acac)₂ | - | - | 37 | 20 |
| 6 | Ni(acac)₂ | L1 | - | 35 | 18 |
| 7 | Ni(acac)₂ | L5 | - | 28 | 9 |
| 8 | Ni(acac)₂ | L1 | B(OH)₃ | 18 | 5 |
| 9 | Ni(acac)₂ | - | B(OH)₃ | 17 | 6 |
| 10 | Ni(acac)₂ | - | MeB(OH)₂ | 39 | 20 |
| 11 | Ni(dpm)₂ | - | - | 36 | 19 |
| 12 | Ni(hfac)₂ | - | - | 30 | 4 |
| 13 | Ni(acac)₂ | - | 2,4-pentane-dione (1.0 equiv.) | 38 | 36 |
| 14 | - | - | 2,4-pentane-dione (1.0 equiv.) | 14 | 0 |

Standard reaction conditions: 0.1 mmol **1a**, 1.0 μmol Ir-F, 5.0 μmol catalyst, 5.0 μmol ligand, 0.6 mmol dicumyl peroxide, 0.5 or 1.0 equiv. additive, MeCN, blue LED irradiation with cooling fan, 25 °C. Yields were determined by ¹H NMR spectroscopy.
[a]Di-tert-butyl peroxide was used in place of dicumyl peroxide, in TFE instead of MeCN.

Ir-F

L1 (4,4',4''-tBu-tpy)

L2

L3 (TPA)

L4

L5 (KTp*)

R = tBu: Ni(dpm)₂
R = CF₃: Ni(hfac)₂

methylation of structurally complex biologically active compounds, and occurs with high selectivity for tertiary C(sp³)−H bonds over other alkyl C−H bonds. The site-selective introduction of methyl groups onto the unactivated C(sp³)−H bonds in active pharmaceutical ingredients and natural products enables the exploration of the "magic methyl" effect in drug discovery.

## Results and discussion

### Development of the undirected methylation of unactivated tertiary C(sp³)−H bonds

Our reaction design was based on photochemical homolysis of the O−O bond in alkyl peroxides[31,32] by triplet energy-transfer from a photosensitizer, (Ir[dF(CF₃)ppy]₂(dtbpy))PF₆ (Ir-F)[33]. Increasingly, this photolytic approach has been used to activate peroxides to generate alkoxyl radicals for C−H functionalizations at milder temperatures than those of the direct thermal, homolytic cleavage of the O−O bond[17,34,35]. The efficiency of C−H bond cleavage by hydrogen atom transfer (HAT) under these photochemical conditions is enhanced over that of thermolysis (>90 °C)[25,26,36–38] because bimolecular HAT outcompetes the relative rate of unimolecular β-methyl-scission at lower temperatures[39,40]. Homolytic bond-dissociation energies (BDE)

of 93−96 kcal mol⁻¹ for tertiary C−H bonds cause these C−H bonds to undergo HAT with alkoxy radicals faster than primary and secondary C−H bonds (BDE > 98 kcal mol⁻¹)[41,42]. The resulting tertiary radical and a Me• generated from an alkoxy radical could then undergo a radical-radical coupling mediated by a catalyst, forging a bond between a quaternary carbon and a methyl group in the product.

We commenced our studies to achieve this reaction by this design by evaluating a series of nickel catalysts and ligands, in combination with Ir-F and dicumyl peroxide under blue LED irradiation. Published conditions[17] delivered the methylated product **2a** from the methylation of isoamyl benzoate **1a** in less than 5% yields (Table 1, entries 1−4). While most of our initial experiments (full details in Supplementary Section 3.1) yielded trace or no product, reactions with Ni(acac)₂ as catalyst gave an appreciable yield (20%) of **2a** (Table 1, entries 5). This observation is consistent with recent reports of C(sp³)−C(sp³) couplings between two sterically distinct carbon radicals in which reactions with Ni(acac)₂ as catalyst gave the highest yields[16,43–48]. The addition of an exogenous tripyridine (**L1**) or KTp* (**L5**) to serve as ligand did not improve the yield (Table 1, entries 6−7), and Lewis acid additives or other nickel(II) acetylacetonate complexes were either detrimental or did not increase the yield (Table 1, entries 8−12). An

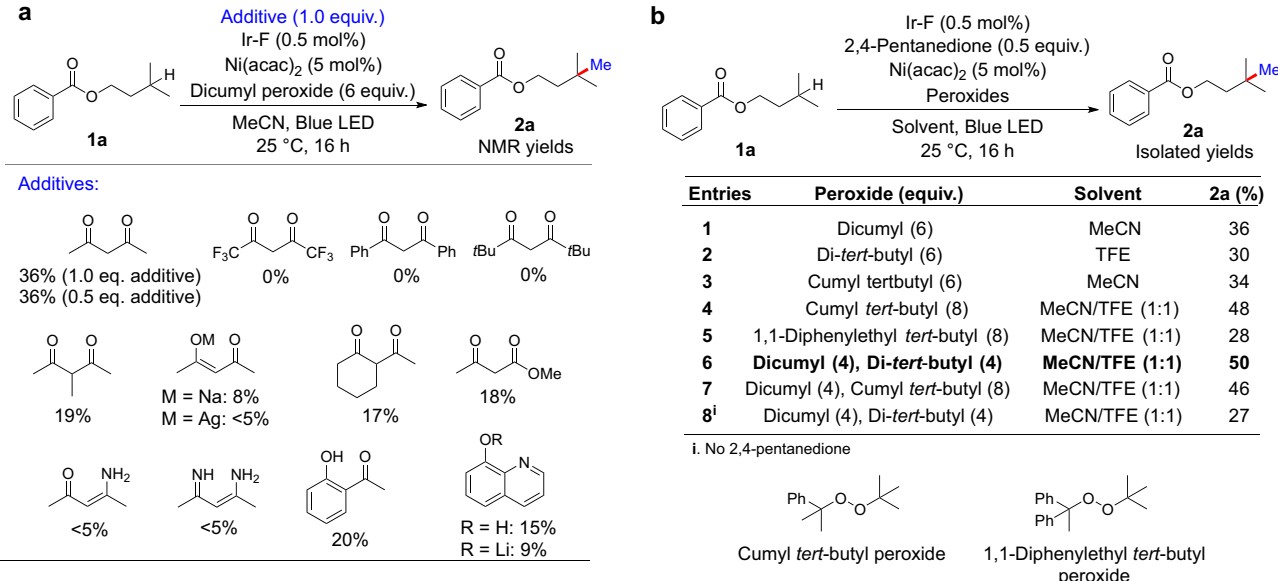

**Fig. 2 | Increase in catalyst turnover with added 2,4-pentanedione. a** Investigations of the effects of various bidentate derivatives. 2,4-Pentanedione led to an increase in yield while the precursors to related bidentate, anionic ligands did not increase the yield. The installed methyl groups are shown in blue. **b** The combination of 50 mol% 2,4-pentanedione with a mixture of dicumyl peroxide and di-*tert*-butyl peroxide gave the highest yield for the undirected methylation of model substrate **1a**. Ir-F = (Ir[dF(CF$_3$)ppy]$_2$(dtbpy))PF$_6$; acac = acetylacetonate; TFE = 2,2,2-trifluoroethanol.

extensive evaluation of various reaction parameters, including catalysts[43–50], solvents, oxidants, and photosensitisers did not increase the yield above 20% (full details in Supplementary Sections 3.1–3.6). Furthermore, control experiments in which the reaction was spiked with the three defined, isolable products (**2a**, α-cumyl alcohol, or acetophenone) demonstrate a lack of inhibition of the reaction by these compounds (Supplementary Information Section 6.1).

However, conducting the methylation with 1 equiv. 2,4-pentanedione led to an increase in yield (Table 1, entry 13). Various other bidentate, anionic additives were tested, but only 2,4-pentanedione led to an increased turnover of the reaction, and the amount of this additive could be reduced to 0.5 equiv. without impacting the yield (Fig. 2a and Supplementary Information Section 3.8). With this finding, we applied some of the previously explored conditions that gave high conversions of **1a** but low yields of **2a**, including the use of unsymmetrical peroxides (Fig. 2b, entries 3–5). The conditions that we eventually selected comprise the combination of dicumyl peroxide and di-*tert*-butyl peroxide in a solvent mixture of TFE and MeCN (Fig. 2b, entry 6). β-Methyl-scission from the alkoxy radicals derived from dicumyl peroxide is faster than from the alkoxy radicals derived from di-tert-butyl peroxide. For the methylation reaction to proceed in high yields, hydrogen atom transfer (HAT) and generation of Me• by β-methyl-scission of alkoxy radicals must occur with similar rates. A combination of the two peroxides can lead to these similar rates, therefore allowing the methylation to proceed in higher yields.

## Substrate scope of methylation

Having identified conditions for the undirected methylation of model substrate **1a**, we evaluated reactions with a series of functional groups on the aryl rings of 4-methylpentyl aryl esters. Substrates bearing electron-withdrawing and electron-donating groups, such as cyano, trifluoromethyl, and methoxy groups, all underwent methylation smoothly with full selectivity for the tertiary C(sp$^3$)−H bonds (Fig. 3, **2a**−**d**). In addition, aryl halides and an aryl pinacol boronic ester (Fig. 3, **2e**−**h**) were well tolerated, underscoring the mildness of this protocol. No reactivity was observed in the presence of redox-sensitive or potentially metal-chelating functionalities, such as a

nitro group (Fig. 3, **2i** or a free amide (Fig. 3, **2j**). This transformation also occurred with substrates containing a variety of pharmaceutically relevant heterocycles, including isoxazole, thiazole, thiophene, and Boc-protected indole (Fig. 3, **2k**−**n**). Heteroarenes containing a basic nitrogen atom, such as pyridines and quinolines, which often poison metal catalysts, were both tolerated and did not undergo methylation by a Minisci-type *ortho*-alkylation, although the methylated products were obtained in lower yields than they were for other classes of heteroarenes (Fig. 3, **2o**−**q**). A naphthyl-substituted ester also formed the product in good yield (Fig. 3, **2r**). We also investigated the methylation of cyclic hydrocarbons. For a comparison between the methylation of these compounds with our protocol and with prior methods, see Section 6.8 of the Supplementary Information.

To determine whether this reaction is applicable to the late-stage methylation of more densely functionalized molecules, we conducted the undirected methylation of a series of biologically active molecules and natural products. A protected L-leucine derivative underwent methylation with good reactivity, allowing access to a non-natural amino acid (Fig. 3, **2s**). Furthermore, the undirected methylation also occurred in a similar fashion for this L-leucine derivative on a 5 mmol scale (1.38 g of **1s**) to deliver the product in 51% yield. While the unprotected acid of L-leucine formed a lower yield than others, this reaction does demonstrate that the reaction can occur in the presence of a free carboxylic acid (Fig. 3, **2t**). A benzoylated derivative of D-Leucinol also underwent methylation under our condition (Fig. 3, **2u**). Moreover, a Leu-Glu dipeptide underwent methylation at the tertiary C(sp$^3$)−H bond (Fig. 3, **2v**), indicating a potential utility of this method for the selective modification of leucine residues in peptides.

The methylation also occurred with a series of derivatives of biologically active molecules. For example, the methylation occurred smoothly with a derivative of sulbactam, a β-lactamase inhibitor (Fig. 3, **2w**). The 4-methylpentyl esters of niflumic acid and fenofibrate, as well as a N−2-methylbutyl derivative of thalidomide, (Fig. 3, **2x**−**z**), all underwent methylation selectively at the tertiary C(sp$^3$)−H bonds. It is worth mentioning that the 2-aminopyridine moiety in **2x** did not

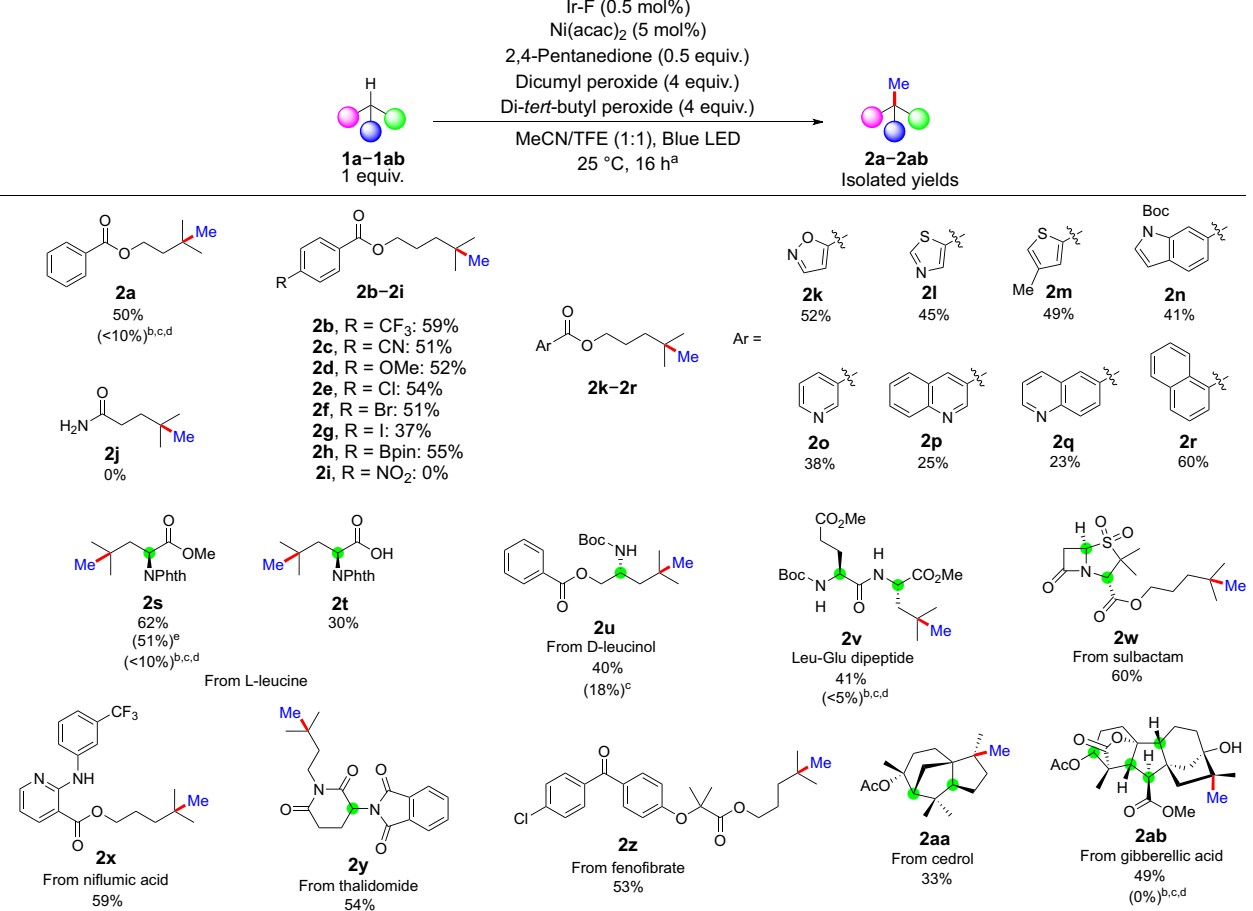

**Fig. 3 | Scope of alkyl substrates that undergo methylation of unactivated tertiary C(sp³)−H bonds.** ªStandard reaction conditions: 0.2 mmol substrate, 1.0 μmol Ir-F, 10.0 μmol Ni(acac)₂, 0.1 mmol 2,4-pentanedione, 0.8 mmol dicumyl peroxide, 0.8 mmol di-*tert*-butyl peroxide, 1:1 MeCN/TFE, blue LED irradiation with cooling fan, 25 °C, isolated yields. The installed methyl groups are shown in blue; the unfunctionalised tertiary C−H bonds are highlighted in green. ᵇPublished conditions (i): 1 mol% Ir-F, 4 mol% NiCl₂•diglyme, 4 mol% 4,4′,4″-ᵗBu-tpy, 1 equiv. substrate, 0.5 equiv. B(OH)₃, 6 equiv. dicumyl peroxide, MeCN, blue LED irradiation with cooling fan, 25 °C, yields by ¹H NMR spectroscopy. ᶜPublished conditions (ii): 1 mol% Ir-F, 4 mol% NiCl₂•diglyme, 4 mol% 4,4′,4″-ᵗBu-tpy, 1 equiv. substrate, 6

equiv. di-*tert*-butyl peroxide, TFE, blue LED irradiation with cooling fan, 25 °C, yields by ¹H NMR spectroscopy. ᵈPublished conditions (iii): 1 mol% Ir-F, 4 mol% NiCl₂•diglyme, 4 mol% TPA, 1 equiv. substrate, 0.5 equiv. MeB(OH)₂, 6 equiv. dicumyl peroxide, MeCN, blue LED irradiation with cooling fan, 25 °C, yields by ¹H NMR spectroscopy. ᵉ5.0 mmol substrate, 25.0 μmol Ir-F, 250.0 μmol Ni(acac)₂, 2.5 mmol 2,4-pentanedione, 20.0 mmol dicumyl peroxide, 20.0 mmol di-*tert*-butyl peroxide, 1:1 MeCN/TFE, blue LED irradiation with cooling fan, 25 °C, isolated yield. Ir-F = (Ir[dF(CF₃)ppy]₂(dtbpy))PF₆; acac = acetylacetonate; TFE = 2,2,2-trifluoroethanol.

interfere with this transformation. The acetyl ester of cedrol, a sesquiterpene alcohol found in cedar oil, underwent methylation exclusively at the more sterically accessible C(sp³)−H bond among the three tertiary C(sp³)−H bonds (Fig. 3, **2aa**). A derivative of the plant growth hormone gibberellic acid, a pentacyclic diterpene, underwent methylation selectively at a single C(sp³)−H bond among five tertiary C(sp³)−H bonds (Fig. 3, **2ab**). The four other tertiary C(sp³)−H bonds did not react because they are sterically encumbered or proximal to electron-withdrawing functionalities. The results of this method were compared to those from conditions published for methylations of activated C(sp³)−H bonds located α to heteroatoms or at benzylic positions. For representative examples **2a, 2s, 2v**, and **2ab** in Fig. 3, the published procedures gave the methylated product in less than 10% yield, and, in the hands of multiple researchers in our group, 18% for **2u**[17]. Overall, in comparison with published conditions, our reaction leads to the methylation of unactivated C(sp³)−H bonds with greater efficiency, exclusive site selectivity, and applicability to the methylation of complex biologically active molecules.

Activated C(sp³)−H bonds α to a heteroatom have BDEs below 90 kcal/mol, due to the stabilization of the corresponding carbon radical by the neighboring heteroatom[40–42]. This effect causes C(sp³)−H

bonds α to a heteroatom to undergo HAT faster than unactivated C(sp³)−H bonds[16–22]. We examined the possibility of methylating unactivated C(sp³)−H bonds in the presence of C(sp³)−H bonds α to a nitrogen atom. Because alkyl amines are prevalent in drug molecules, the selective methylation of unactivated C(sp³)−H bonds in the presence of alkyl amines would increase the range of substrates amenable to this methodology. Leveraging a complexation with BF₃ of the nitrogen in secondary alkyl amines[51], we conducted methylation of the tertiary C(sp³)−H bond in the BF₃ adduct of 4-methyl piperidine (Fig. 4, **5a**). The methylation occurred cleanly with full selectivity at the unactivated tertiary C(sp³)−H bond. The four C(sp³)−H bonds α to the nitrogen were not functionalized, due to the inductive effect of the borane coordination that renders the adjacent C(sp³)−H bonds electron-deficient[52,53]. In contrast, protecting groups such as Boc and Bz groups on the nitrogen of 4-methyl piperidine led to complex mixtures under the conditions for the methylation of C(sp³)−H bonds, whereas the hydrochloride salt of 4-methyl piperidine gave less than 5% conversion (Supplementary Section 3.11). The selective methylation of tertiary C(sp³)−H bonds also occurred in amine−BF₃ complexes of benzyl amine and isoindoline (Fig. 4, **5b, c**). Decomplexation of the BF₃ adduct after methylation occurred with cesium fluoride (Fig. 4, **6**).

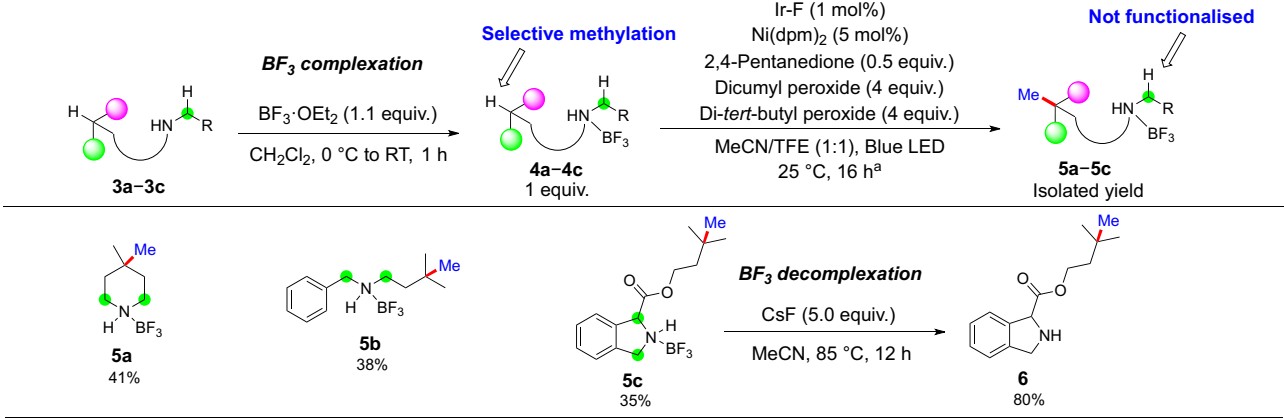

**Fig. 4 | Scope of substrates containing BF₃-protected amines that undergo methylation at unactivated tertiary C(sp³)−H bonds.** [a]Standard reaction conditions: 0.2 mmol amine−BF₃ substrate, 1.0 µmol Ir-F, 10.0 µmol Ni(dpm)₂, 0.1 mmol 2,4-pentanedione, 0.8 mmol dicumyl peroxide, 0.8 mmol di-*tert*-butyl peroxide, 1:1 MeCN/TFE, blue LED irradiation with cooling fan, 25 °C, isolated yields. The installed methyl groups are shown in blue; the activated C(sp³)−H bonds that are not methylated are highlighted in green. Ir-F = (Ir[dF(CF₃)ppy]₂(dtbpy))PF₆; dpm = 2,2,6,6-tetramethyl-3,5-heptanedionate; TFE = 2,2,2-trifluoroethanol.

## Mechanistic investigations

To gain insight into the mechanism of this methylation and higher catalytic turnover of Ni(acac)₂ when conducted with added 2,4-pentanedione, we first investigated the effect of varying the amounts of the components in the reaction without added 2,4-pentanedione. With a constant loading of Ni(acac)₂ of 5 mol%, we found that increasing the loading of Ir-F from 1 mol% to 2 mol% led to a decreased conversion of dicumyl peroxide (Fig. 5a, entries 1 and 2). This observation suggested that Ir-F catalyzes its own deactivation, presumably by accelerating the rate of the production of Me•, which adds to the aromatic ligands of Ir-F and deactivates the photosensitiser in the absence of Ni(acac)₂. The conversion of dicumyl peroxide in the presence of 2 mol% Ir-F and 10 mol% Ni(acac)₂ was comparable to that in the presence of 1 mol% Ir-F and 5 mol% Ni(acac)₂ (Fig. 5a, entry 3), indicating that Ni(acac)₂ competes with Ir-F for Me•. Consistent with this hypothesis, the conversion of dicumyl peroxide was just 14% in the absence of Ni(acac)₂ (Fig. 5a, entry 4). Thus, the presence of the Ni(II) catalyst appears to inhibit the deactivation of Ir-F, leading to a corresponding increase in the conversion of the peroxide. More detailed studies that indicate this interplay between Ir-F and Ni(acac)₂ can be found in Supplementary Information Section 6.9.

To understand the mode of deactivation of the photosensitiser, we studied the fate of Ir-F by subjecting aliquots of a reaction (Fig. 5a, entry 1) to LC-MS analysis after 4 h and 16 h, respectively (full details in Supplementary Information Section 6.3). After 4 h, an *m/z* matching the molecular weight of the unmodified Ir-F complex ([M] = 977) was identified in the LC-MS chromatogram (Fig. 5a, bottom left). In contrast, after 16 h, the m/z of the major Ir species was [M + 14] and [M + 28], signifying monomethylation and dimethylation of Ir-F (Fig. 5a, bottom right). The same set of *m/z* was observed when the reaction was performed in the absence of nickel catalysts[17]. These findings indicate that methylations of the aromatic ligands of Ir-F likely lead to deactivation of the photosensitiser in the absence of Ni(acac)₂ or after Ni(acac)₂ is consumed. Consistent with this assertion, the deactivation of Ir-F in the absence of Ni(acac)₂ was accompanied by a change in color of the reaction mixture from the usual bright yellow to deep orange.

We also determined the changes to dicumyl peroxide, substrate **1a**, and product **2a** as a function of time in the absence of added 2,4-pentanedione (Fig. 5b). The formation of **2a** reached a maximum of 20% after ~4 h. In contrast, the conversion of dicumyl peroxide and **1a** continued after this time. This observation implies that the Ni(acac)₂ or the catalyst formed from it is fully deactivated after about 4 h,

preventing further methyl−C(sp³) coupling, but the peroxide continues to react in the presence of Ir-F to generate cumyl alcohol and acetophenone, and, presumably, unproductive formation of Me•. Studies on the fate of the acetylacetonate ligand suggest that it undergoes methylation to form 3-methyl-2,4-pentanedione. The addition of 2,4-pentanedione potentially restores the active form of the Ni catalyst, thereby leading to higher turnovers. Consistent with this hypothesis, a control experiment conducted with 2,4-pentanedione and without **1a** under our photochemical condition resulted in the formation of 3-methyl-2,4-pentanedione (15% yield, Fig. 6 and Supplementary Information Section 6.5). This hypothesis is also consistent with reports of Cu-catalyzed methylations of 1,3-dicarbonyl derivatives[54,55].

In conclusion, we have developed an undirected, site-selective methylation of unactivated tertiary C(sp³)−H bonds that is applicable to the modification of peptides, terpenes, and active pharmaceutical ingredients. One key to this transformation is a simple nickel(II) acetylacetonate complex whose lifetime is enhanced by added 2,4-pentanedione. The methodology can be applied to the late-stage methylation of complex organic molecules that contain multiple alkyl C−H bonds with high selectivity for tertiary C(sp³)−H bonds and a high compatibility with a wide array of functional groups. As a result, this procedure provides a direct route to methylated derivatives of biologically active compounds and natural products. Mechanistic studies uncovered a delicate interplay between the nickel catalyst and the Ir photosensitiser and revealed a potential regeneration of the catalytically active nickel complex mediated by 2,4-pentanedione. We anticipate the results of this study to inspire the future development of the methylation of unactivated secondary or primary C(sp³)−H bonds.

## Methods

### General procedure for the methylation of unactivated C(sp³)−H bonds

Ir-F (1.2 mg, 0.5 mol%, 1.0 µmol), Ni(acac)₂ (2.6 mg, 5 mol%, 10.0 µmol), and dicumyl peroxide (0.800 mmol, 4 equiv, 216 mg) were weighed into a vial charged with a magnetic stirrer. A pre-formed 1:1 mixture of degassed MeCN and TFE (0.2 mL) were added, followed by the substrate (0.200 mmol, 1.00 equiv), di-*tert*-butyl peroxide (0.800 mmol, 4 equiv, 146 µL), and 2,4-pentanedione (10.0 µmol, 0.5 equiv, 10 µL). The vial was sealed, and the headspace of the vial was flushed with a stream of N₂ for 1 min. The mixture was then stirred and irradiated under a blue LED Kessil lamp for 16 h with a cooling fan maintaining the temperature at 25 °C. The reaction mixture was concentrated under

**a**

1a (1 equiv.)
+
Dicumyl peroxide
(6 equiv.)

Ir-F (**x** mol%)
Ni(acac)$_2$ (**y** mol%)
———————————→
MeCN, Blue LED
25 °C, 16 h

2a

| Entries | x | y | Peroxide conv. (%) | 2a (%) |
|---|---|---|---|---|
| **1** | 1 | 5 | 69 | 20 |
| **2** | 2 | 5 | 35 | 12 |
| **3** | 2 | 10 | 66 | 20 |
| **4** | 1 | - | 14 | 0 |

Entry **1** (4 hours)
Major species: [M]

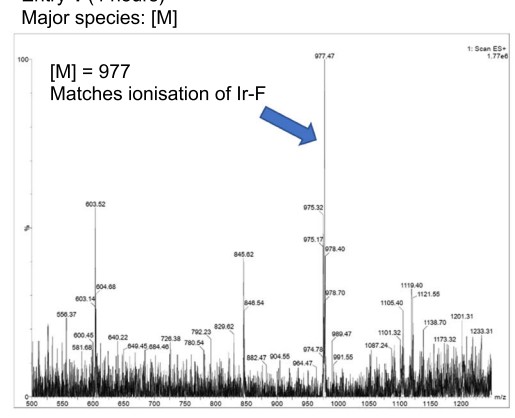

[M] = 977
Matches ionisation of Ir-F

Entry **1** (16 hours)
Major species: [M+14], [M+28]

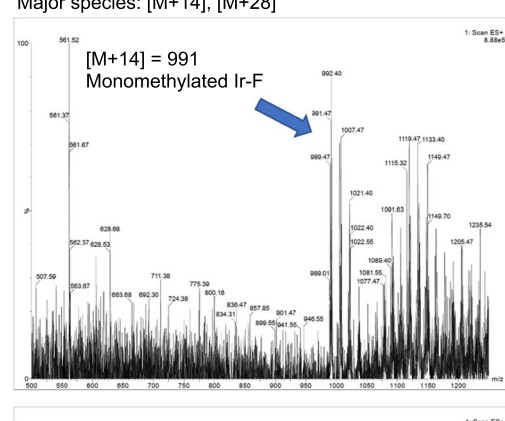

[M+14] = 991
Monomethylated Ir-F

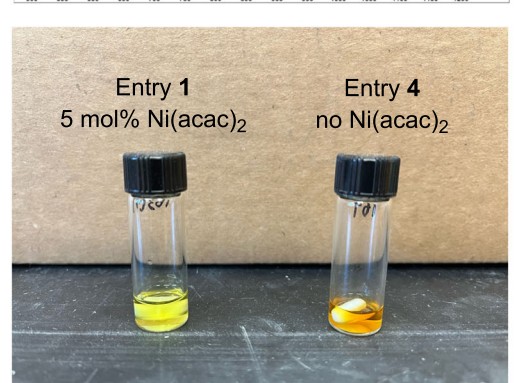

Entry **1**
5 mol% Ni(acac)$_2$          Entry **4**
no Ni(acac)$_2$

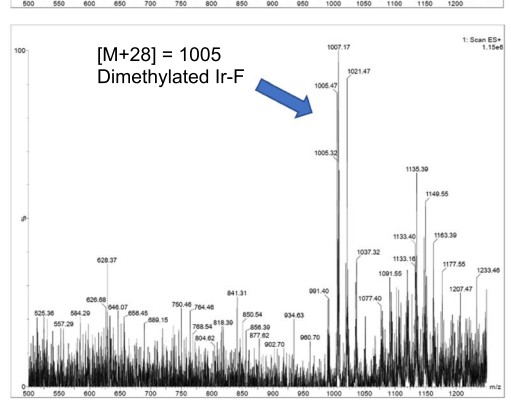

[M+28] = 1005
Dimethylated Ir-F

**b**

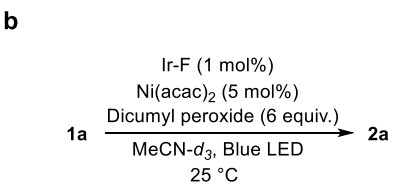

1a

Ir-F (1 mol%)
Ni(acac)$_2$ (5 mol%)
Dicumyl peroxide (6 equiv.)
———————————→
MeCN-$d_3$, Blue LED
25 °C

2a

| Time (h) | Peroxide (%) | 1a (%) | 2a (%) |
|---|---|---|---|
| 4 | 57 | 76 | 19 |
| 6 | 44 | 68 | 20 |
| 16 | 30 | 62 | 20 |

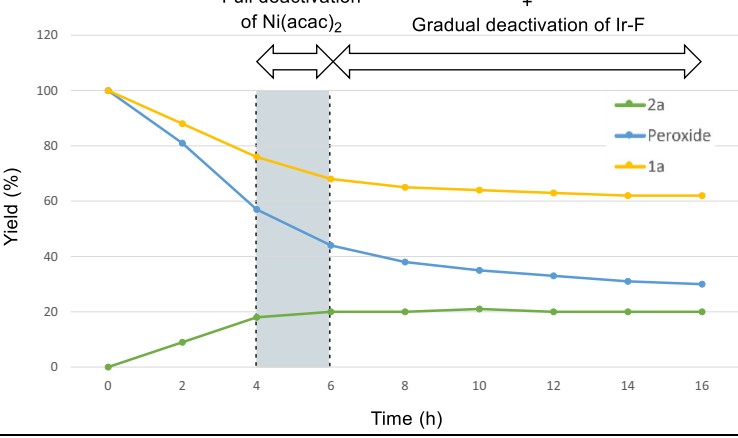

**Fig. 5 | Mechanistic investigations to identify features controlling catalyst activity. a** Conversions of peroxides with varying loadings of Ir-F and Ni(acac)$_2$. A higher loading of Ir-F leads to more rapid inactivation of the Ir-F toward activation of the peroxides, while the presence of Ni(acac)$_2$ increases the lifetime of Ir-F. Investigations of the fate of Ir-F after the reaction, based on the mass spectra of the major peaks in the UV-vis trace of the LC chromatograms after the reactions. Monomethylated and dimethylated species of Ir-F were detected in the absence of Ni(acac)$_2$. The picture depicts the difference in color of the two reaction mixtures at the end of the reaction with and without Ni(acac)$_2$. **b** Monitoring the quantitative changes of **1a**, **2a**, and dicumyl peroxide over time. The termination of the formation of **2a** after ~4 h, paired with the continued consumption of dicumyl peroxide and **1a** after 4 h, suggest that the full deactivation of Ni(acac)$_2$ occurs before the full deactivation of Ir-F. Ir-F = (Ir[dF(CF$_3$)ppy]$_2$(dtbpy))PF$_6$; acac = acetylacetonate.

reduced pressure and filtered through a short silica plug, followed by rinsing the plug with CH$_2$Cl$_2$. Volatile materials were evaporated, and 1,3,5-trimethoxybenzene (internal standard) was added to the crude mixture. Crude $^1$H NMR spectroscopy was performed in CDCl$_3$ to quantify the conversion and the yield. After recovering the analytical sample and concentrating under reduced pressure, the crude residue was purified by flash column chromatography (silica and C18 reverse phase) to give the methylated product.

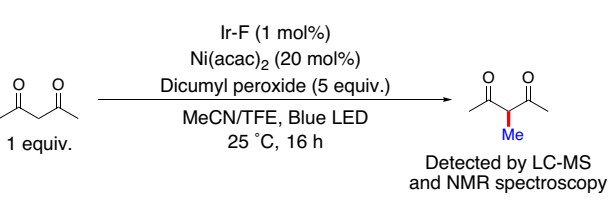
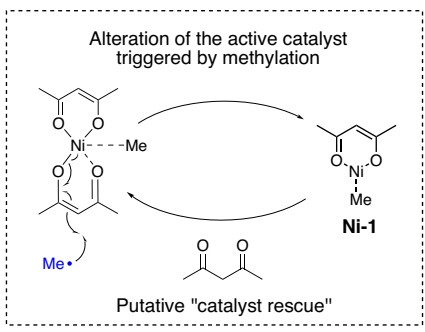

**Fig. 6 | Possible deactivation of the Ni catalyst triggered by methylation and ligand-mediated regeneration of the active catalytic species.** Detection of 3-methyl-2,4-pentanedione under our photochemical condition. The reaction of Ni(acac)₂ with a methyl radical is a potential mode of catalyst deactivation. The regeneration of the catalytically active Ni(acac)₂ in the presence of exogenous 2,4-pentanedione potentially contributes to the increased turnover of Ni in the methylation of C(sp³)–H bonds catalyzed by Ni(acac)₂. Ir-F = (Ir[dF(CF₃) ppy]₂(dtbpy))PF₆; acac = acetylacetonate; TFE = 2,2,2-trifluoroethanol.

## Data availability

Complete experimental procedures and compound characterization data are available in the Supplementary Information; any other data are available from the authors on request.

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

## Acknowledgements

This work was supported by the NIGMS of the NIH under R35GM130387. J.-F.T. is supported by a Swiss National Science Foundation (SNSF) Postdoc.Mobility Fellowship (P500PN_202713). We thank Drs. Hasan Celik, Raynald Giovine, and Pines Magnetic Resonance Center's Core NMR Facility (PMRC Core) for spectroscopic assistance. The instruments used in this work were in part supported by NIH S10OD024998. We thank the QB3/Chemistry Mass Spectrometry Facility and the Catalysis Center at UC Berkeley for assistance with mass spectrometry.

## Author contributions

J.F.T. and J.F.H. conceived and designed the project. J.F.T. and Y.C.K. performed the experimental studies. J.F.T. and Y.C.K. analyzed and interpreted experimental data. J.F.T. and J.F.H. wrote the manuscript.

## Competing interests

The authors declare no competing interests.
