## [Peer Review File · Nature Communications]

Catalytic Undirected Methylation of Unactivated C(sp³)-H Bonds Suitable for Complex MoleculesREVIEWER COMMENTS

Reviewer #1 (Remarks to the Author):

In this manuscript, Hartwig and coworkers present an approach to the catalytic undirected methylation of unactivated C(sp³)-H bonds employing nickel/photoredox catalysis. The authors have successfully demonstrated the method's broad functional group tolerance and its predilection for tertiary C-H bonds. This feature holds particular significance for the late-stage functionalization of intricate molecules including terpenes, peptides, and pharmaceuticals, which inherently possess a multitude of alkyl C-H bonds. Despite the prior introduction of peroxides as bifunctional reagents by the Stahl group (referenced as ref. 17), Hartwig's group has made a commendable stride in extending the scope of methylation from α -C(sp³)-H bonds to their unactivated counterparts. This advancement has noteworthy implications for the synthesis of methylated analogs of therapeutic agents. A further point of interest is the identification of an anionic additive, specifically 2,4-pentanedione, which ostensibly enhances the nickel catalyst turnover. This discovery could serve as a design element for future research into SH₂-type catalytic reactions. On the strength of these findings, I endorse the publication of this manuscript in Nature Communications.

However, some specific points should be addressed:

1) In Figure 1c, substrates from Li's study, namely decahydronaphthalene and hexane, warrant investigation under the current study's conditions. The site selectivity observed seems to be governed by the substrate's intrinsic electronic properties rather than the influence of HAT reagents or nickel catalysts. This raises the possibility of observing non-selective methylation with these substrates as well. To maintain analytical integrity, a revision of the corresponding narrative is recommended to enable a balanced comparison.

2) The manuscript notes a marked improvement in yield with the combined use of dicumyl peroxide and di-tert-butyl peroxide. An insightful commentary on this observation would be beneficial for readers seeking a deeper understanding of the underlying chemistry.

3) There is a discrepancy in the yield of Compound 2u as reported by Stahl's group (53%, ref. 17) and the yield obtained by the authors (18%). This significant difference calls for clarification. Is there a reproducibility issue that needs to be addressed?

4) Given the stoichiometric use of peroxides, it is prudent to incorporate explicit safety guidelines within the SI to ensure safe handling and experimental procedures.

5) In Figure 1b, the spelling error "Macmillan" should be corrected to "MacMillan" to maintain the accuracy of the text.

Reviewer #2 (Remarks to the Author):

In this study, Hartwig and Tan depicted a Photo/Ni-catalyzed methylation of tertiary alkyl C-H bonds, affording methylated all carbon quaternary centers in high regioselectivity. DTBP and dicumyl peroxide serve as the methyl sources upon homolysis/1e reduction of the weak O-O bonds followed by C-C fragmentation to generate methyl radical. The substrate scope of the present work is broad, which is in particular suitable for terminal dimethyl ones. Competitive N- α -CH bonds of secondary amines were deactivated with a N-BF₃-procting strategy. It should be noted that coupling yields of the present methylation process are generally moderate, and the concept of using peroxides as for C(sp³)-H methylation under Photo/Ni conditions has been unveiled by Stahl. However, construction of methylated all carbon quaternary centers for unactivated tertiary C(sp³)-H bonds remain a challenge. This work demonstrates excellent site selectivity, and the coupling efficiency can be boosted with 2,4-pentanedione, likely due to activating Ni so as to inhibit methylation/deactivation of Ir-F. These combined new features warrant its publication in Nat. Commun, provided the following minor notes are addressed.

1. Because two peroxides are used in the coupling process, line 4 of the first paragraph in the mechanistic section, peroxide can be specified as dicumyl peroxide.

2. Scheme 1e, the catalytic cycles are not easy to understand, in particular Ni(n+1)Me to Ni(n)Me.

3. It is likely that reduction of peroxide may involve an inner sphere O-M coordination process. I suggest the authors compare the kinetics of consumption of peroxide for a mixture of dicumyl peroxide with Ir-F (1%) and a mixture of dicumyl peroxide with Ir-F (1%) and Ni(acac)₂, by monitoring the reaction for the first few hours. Perhaps, the kinetic profiles may suggest that less sterically hindered Ni species (likely Ni(I)) at higher concentration govern the O-O bond reduction process. For Ni(I) reduction, see: *Angew. Chem.* (2022, e202201662).

4. A control without Ni can be included in Table 1.

Reviewer #3 (Remarks to the Author):

In this manuscript, Prof. Hartwig and coworkers reported an undirected, site-selective methylation of unactivated tertiary C(sp³)-H bonds. This fills a gap in current synthetic capabilities, as existing methods predominantly focus on the methylation of C(sp³)-H bonds situated α to heteroatoms or in benzylic positions. Central to this transformation is the use of a nickel(II) acetylacetonate complex, whose lifetime is enhanced by added 2,4-pentanedione. This enhancement plays a pivotal role in augmenting the turnover frequency of nickel during the methylation of C(sp³)-H bonds. This work holds promise for expediting the synthesis of methylated drug analogs, thereby enabling a more efficient exploration of the "magic methyl" effect, which often enhances pharmacological properties. While acknowledging a noteworthy parallel between this method – integrating photochemically induced peroxides with a nickel(II) complex – and the preceding study by the Stahl group (Reference 17). The facile introduction of an auxiliary ligand into the system addresses the challenge of undirected, site-selective methylation of unactivated tertiary C(sp³)-H bonds, which is a notable advancement. I am in favor of its publication in *Nature Communications*, conditional upon satisfactorily addressing the following raised concerns.

(1) The concentration of the reaction mixture plays a crucial role in modulating both the rate of β -methyl cleavage and the hydrogen atom transfer (HAT) steps involved in the methylation process. Consequently, it is imperative to consider reaction concentration as a key variable during the optimization of the experimental conditions. Furthermore, an investigation into the effect of temperature variation on the methylation efficiency would

augment the understanding of the reaction dynamics. To address this, inclusion of additional experimental data points, reflecting increased and decreased temperatures, in the results table is advisable.

(2) Are there any observations of secondary carbon methylation products in substrates that simultaneously contain secondary and tertiary carbons, particularly when the secondary carbon is located α to a heteroatom? In addition, the majority of these reactions yield outputs in the moderate to lower range. What are the main factors affecting these yields, and is it possible to further enhance them?

(3) The author compares the outcomes of some reactions with those reported under previously documented reaction conditions, such as, in the synthesis of compounds 2s, 2u, 2v, and 2ab, where the experimental exceed those reported under established reaction conditions. Could you offer a reasonable explanation for this discrepancy?

(4) Comparison through entries 1-5 in Figure 2b, the condition that a mixture of dicumyl peroxide and cumyl tert-butyl peroxide in a solvent mixture of TFE and MeCN should be added in the manuscript.

(5) Please carefully check the main text and Supplementary Information, there are a number of errors and inconsistencies: a) The yield of compound 2x is inconsistent between the manuscript and ESI. b) In the optimization of the reaction conditions, the yield of compound 2a is inconsistent between the manuscript (Table 1, entry 6) and ESI (Section 3.1, entry 7). c) Condition b should be noted in Section 3.10 in the ESI. d) The HRMS analysis is highly biased for compound 4c. e) Reference punctuation formatting symbols should be standardized.

(6) The following compounds are not pure, and therefore need to be re-purified, such as, compounds 1q, 1ab, 4b, 2a, 2z.

Point-to-point responses to reviewers' comments

Reviewer #1 (Remarks to the Author):

In this manuscript, Hartwig and coworkers present an approach to the catalytic undirected methylation of unactivated C(sp³)-H bonds employing nickel/photoredox catalysis. The authors have successfully demonstrated the method's broad functional group tolerance and its predilection for tertiary C-H bonds. This feature holds particular significance for the late-stage functionalization of intricate molecules including terpenes, peptides, and pharmaceuticals, which inherently possess a multitude of alkyl C-H bonds. Despite the prior introduction of peroxides as bifunctional reagents by the Stahl group (referenced as ref. 17), Hartwig's group has made a commendable stride in extending the scope of methylation from α-C(sp³)-H bonds to their unactivated counterparts. This advancement has noteworthy implications for the synthesis of methylated analogs of therapeutic agents. A further point of interest is the identification of an anionic additive, specifically 2,4-pentanedione, which ostensibly enhances the nickel catalyst turnover. This discovery could serve as a design element for future research into SH2-type catalytic reactions. On the strength of these findings, I endorse the publication of this manuscript in Nature Communications.

However, some specific points should be addressed:

1) In Figure 1c, substrates from Li's study, namely decahydronaphthalene and hexane, warrant investigation under the current study's conditions. The site selectivity observed seems to be governed by the substrate's intrinsic electronic properties rather than the influence of HAT reagents or nickel catalysts. This raises the possibility of observing non-selective methylation with these substrates as well. To maintain analytical integrity, a revision of the corresponding narrative is recommended to enable a balanced comparison.

Authors' response:

We appreciate the reviewer's suggestion. We performed reactions with cyclohexane and *cis*-decalin under the standard reaction conditions. The yield of methylcyclohexane was 0.8% (determined by GC-FID and comparison to an authentic standard, due to the volatility of the product). The yield of *cis*-9-methyldecalin was <5%, while the yield of *trans*-9-methyldecalin was 27% (determined by ¹H NMR spectroscopy of the crude product). The yields of 1-methyl and 2-methyl-*trans*-decalin were <5%.

These results demonstrate that the site selectivity of hydrogen atom abstraction is not solely governed by the substrate's intrinsic electronic properties. The system of Li and coworkers is more reactive toward secondary C–H bonds. They reported a 37% yield for the methylation of cyclohexane, while we observed only trace yield with cyclohexane.

In addition, they reported that decalin underwent methylation in 12% yield at the secondary C–H bonds (3:2 ratio of 1-methyl to 2-methyl-*trans*-decalin, no tertiary C–H methylation observed). In contrast, our protocol gave a 27% yield of *trans*-9-methyldecalin and <5% yield of 1-methyl and 2-methyl-*trans*-decalin. Thus, the methylation of decalin catalyzed by GaN is selective for secondary C–H bonds while our reaction is highly selective for tertiary C–H bonds. While no selectivity can be observed in the methylation of cyclohexane, the low reactivity of that substrate under our reaction conditions also is consistent with a higher rate of functionalization of tertiary C–H bonds than secondary C–H bonds.

The formation of *trans*-9-methyldecalin starting from *cis*-decalin suggests that hydrogen atom abstraction of the tertiary C(*sp*³)–H bonds in *cis*-decalin occurs more rapidly than methylation of the tertiary radical by the nickel catalyst. We hypothesize that epimerization occurs by the tertiary radical abstracting a hydrogen atom from solvent.

We have added these experiments and the results to Section 6.8 of the Supplementary Information. We have also added a sentence in the manuscript that reads: “We also investigated the methylation of cyclic hydrocarbons. For a comparison between the methylation of these compounds with our protocol and with prior methods, see Section 6.8 of the Supplementary Information.”

2) The manuscript notes a marked improvement in yield with the combined use of dicumyl peroxide and di-*tert*-butyl peroxide. An insightful commentary on this observation would be beneficial for readers seeking a deeper understanding of the underlying chemistry.

Authors' response:

We appreciate this constructive feedback. β -Methyl-scission from the alkoxy radicals derived from dicumyl peroxide is faster than that from the alkoxy radicals derived from di-*tert*-butyl peroxide. For the methylation reaction to proceed in high yields, hydrogen atom transfer (HAT) and generation of Me• by β -methyl-scission of alkoxy radicals must occur with similar rates. A combination of the two peroxides can lead to these similar rates and the final methylation process, as long as Ni(acac)₂ remains catalytically active.

A sentence has been included in the manuscript (page 4) for a better understanding. This sentence reads, “ β -Methyl-scission from the alkoxy radicals derived from dicumyl peroxide is faster than from the alkoxy radicals derived from di-*tert*-butyl peroxide. For the methylation reaction to proceed in high yields, hydrogen atom transfer (HAT) and generation of Me• by β -methyl-scission of alkoxy radicals must occur with similar rates. A combination of the two peroxides can lead to these similar rates, therefore allowing the methylation to proceed at higher yields.”

3) There is a discrepancy in the yield of Compound 2u as reported by Stahl's group (53%, ref. 17) and the yield obtained by the authors (18%). This significant difference calls for clarification. Is there a reproducibility issue that needs to be addressed?

Authors' response:

We were unable to reproduce the results reported by Stahl's group. In the hands of multiple researchers in our group, the yields of **2u** that we obtained range from 17%-18%. Furthermore, we have ensured that the reagents used are high quality, including use of newly purchased nickel precursor and ligand.

4) Given the stoichiometric use of peroxides, it is prudent to incorporate explicit safety guidelines within the SI to ensure safe handling and experimental procedures.

Authors' response:

Safety guidelines for the safe handling of peroxides have been included on page 6 of SI. These read,

"Peroxides, peracetates, perbenzoates, and peracids are known to pose explosion risks. Therefore, appropriate care should be taken when handling these reagents. For example, do not store these chemicals in open, partially empty, or transparent containers. Do not allow open flames, other sources of heat or sparks, friction, grinding, or forms of impact near these reagents.

For more comprehensive safety guidelines, please see:

<https://ehs.berkeley.edu/sites/default/files/pecguidelines.pdf>"

5) In Figure 1b, the spelling error "Macmillan" should be corrected to "MacMillan" to maintain the accuracy of the text.

Authors' response:

The spelling error has been corrected.

Reviewer #2 (Remarks to the Author):

In this study, Hartwig and Tan depicted a Photo/Ni-catalyzed methylation of tertiary alkyl C-H bonds, affording methylated all carbon quaternary centers in high regioselectivity. DTBP and dicumyl peroxide serve as the methyl sources upon homolysis/1e reduction of the weak O-O bonds followed by C-C fragmentation to generate methyl radical. The substrate scope of the present work is broad, which is in particular suitable for terminal dimethyl ones. Competitive N- α -CH bonds of secondary amines were deactivated with a N-BF₃-procting strategy. It should be noted that coupling yields of the present methylation process are generally moderate, and the concept of using peroxides as for C(sp³)-H methylation under Photo/Ni conditions has been unveiled by Stahl. However, construction of methylated all carbon quaternary centers for unactivated tertiary C(sp³)-H bonds remain a challenge. This work demonstrates excellent site selectivity, and the coupling efficiency can be boosted with 2,4-pentanedione, likely due to activating Ni so

as to inhibit methylation/deactivation of Ir-F. These combined new features warrant its publication in Nat. Commun, provided the following minor notes are addressed.

1. Because two peroxides are used in the coupling process, line 4 of the first paragraph in the mechanistic section, peroxide can be specified as dicumyl peroxide.

Authors' response:

“The peroxide” has been changed to “dicumyl peroxide”.

2. Scheme 1e, the catalytic cycles are not easy to understand, in particular Ni(n+1)Me to Ni(n)Me.

Authors' response:

Scheme 1e have been modified, using Ni(acac)₂ and its related species for a clearer depiction of the catalytic cycles. The prior and revised Scheme 1e is shown below:

Old scheme:

- e Proposed mechanism of the methylation of unactivated tertiary C(sp³)-H bonds mediated by radical-radical coupling and ancillary-ligand promoted enhancement of catalytic turnover of nickel complex

New scheme:

- e Proposed mechanism of the methylation of unactivated tertiary C(sp³)-H bonds mediated by radical-radical coupling and ancillary-ligand promoted enhancement of catalytic turnover of nickel complex

3. It is likely that reduction of peroxide may involve an inner sphere O-M coordination process. I suggest the authors compare the kinetics of consumption of peroxide for a mixture of dicumyl peroxide with Ir-F (1%) and a mixture of dicumyl peroxide with Ir-F (1%) and Ni(acac)₂, by monitoring the reaction for the first few hours. Perhaps, the

kinetic profiles may suggest that less sterically hindered Ni species (likely Ni(I)) at higher concentration govern the O-O bond reduction process. For Ni(I) reduction, see: *Angew. Chem.* (2022, e202201662).

Authors' response:

As suggested by the reviewer, we performed experiments to determine the consumption of dicumyl peroxide in the presence of the iridium photocatalyst alone and in the presence of both the iridium and nickel catalysts over the course of 4 h. The plots below show that the rate of conversion of dicumyl peroxide and the rate of formation of acetophenone are both greater in the presence of both catalysts (Condition 2) than it is in the presence of the iridium photocatalyst alone (Condition 1).

The table below summarizes the conversion of dicumyl peroxide and the formation of acetophenone at 4 hours under Conditions 1-4.

Condition	Catalysts	DCP remaining (mmol)	Acetophenone (mmol)
Initial	-	0.600	0.000
1	Ir	0.352	0.313
2	Ir and Ni	0.055	0.769
3	Ni	0.332	0.420
4	None	0.349	0.384

The homolysis of dicumyl peroxide under irradiation by 400 nm LEDs is significant (Conditions 4), as 42% of the peroxide is consumed and a 32% yield of acetophenone is observed (via β -scission). The conversion of dicumyl peroxide and formation of acetophenone in the presence of the iridium photocatalyst alone (Condition 1) or the nickel catalyst alone (Condition 3) are comparable to those of the background reaction (Condition 4). In contrast, when both catalysts are present (Condition 2), the conversion of dicumyl peroxide and formation of acetophenone are both greatly increased.

These results are consistent with our mechanistic investigations and support catalyst cooperativity. Neither catalyst alone is competent at promoting the homolysis of dicumyl peroxide, but the combination of the two catalysts leads to increased consumption of the peroxide and formation of acetophenone above the background reaction. The iridium photocatalyst is capable of promoting homolysis of dicumyl peroxide by energy transfer, but it is rapidly deactivated by the methyl radicals generated by β -scission in

the absence of the nickel catalyst. The nickel catalyst prolongs the lifetime of the iridium catalyst by capturing methyl radicals.

These data have been added to the SI, in Section 6.9 (page 69-72). A sentence has also been included in the manuscript (page 8, first paragraph) referring to the addition of these data. This sentence reads, "More detailed studies that indicate this interplay between Ir-F and Ni(acac)₂ can be found in Supplementary Information Section 6.9."

4. A control without Ni can be included in Table 1.

Authors' response:

The control experiment without Ni has been included in Table 1 (entry 14). This entry shows 14% conversion of isoamyl benzoate 1a and 0% formation of methylated product 2a.

Reviewer #3 (Remarks to the Author):

In this manuscript, Prof. Hartwig and coworkers reported an undirected, site-selective methylation of unactivated tertiary C(sp³)-H bonds. This fills a gap in current synthetic capabilities, as existing methods predominantly focus on the methylation of C(sp³)-H bonds situated α to heteroatoms or in benzylic positions. Central to this transformation is the use of a nickel(II) acetylacetonate complex, whose lifetime is enhanced by added 2,4-pentanedione. This enhancement plays a pivotal role in augmenting the turnover frequency of nickel during the methylation of C(sp³)-H bonds. This work holds promise for expediting the synthesis of methylated drug analogs, thereby enabling a more efficient exploration of the "magic methyl" effect, which often enhances pharmacological properties. While acknowledging a noteworthy parallel between this method – integrating photochemically induced peroxides with a nickel(II) complex – and the preceding study by the Stahl group (Reference 17). The facile introduction of an auxiliary ligand into the system addresses the challenge of undirected, site-selective methylation of unactivated tertiary C(sp³)-H bonds, which is a notable advancement. I am in favor of its publication in Nature Communications, conditional upon satisfactorily addressing the following raised concerns.

(1) The concentration of the reaction mixture plays a crucial role in modulating both the rate of β -methyl cleavage and the hydrogen atom transfer (HAT) steps involved in the methylation process. Consequently, it is imperative to consider reaction concentration as a key variable during the optimization of the experimental conditions. Furthermore, an investigation into the effect of temperature variation on the methylation efficiency would augment the understanding of the reaction dynamics. To address this, inclusion of additional experimental data points, reflecting increased and decreased temperatures, in the results table is advisable.

Authors' response:

Entry	X	Y	NMR Yield (%)
1	0.33	25	17
2	0.67	25	19
3	1.0	25	20
4	2.0	25	20
5	1.0	40	24
6	1.0	60	12
7	1.0	80	7

With higher concentrations, the yields were slightly higher. However, the yield from the reaction with a concentration of **1a** at 2.0 M was the same as that from the reaction at 1.0 M.

When the reaction was performed at 40 °C, the yield of product was comparable to that at 25 °C. Further increases in temperature (60 °C, 80 °C) led to significant reductions in yield. All further reactions were performed at room temperature, due to operational ease.

We have included the results from varying the concentration and temperature in Section 3.6, page 30 of SI.

(2) Are there any observations of secondary carbon methylation products in substrates that simultaneously contain secondary and tertiary carbons, particularly when the secondary carbon is located α to a heteroatom? In addition, the majority of these reactions yield outputs in the moderate to lower range. What are the main factors affecting these yields, and is it possible to further enhance them?

Authors' response:

Yes, it is likely that our methylation occurs at a secondary C–H bond α to nitrogen. We have originally conducted our reactions on substrates that contain both a tertiary C–H bond and a secondary C–H bond that is α to a nitrogen (See Section 3.11 of SI). For example, under our conditions, Boc-protected 4-methylpiperidine formed a complex mixture of products that indicated unselective methylation at both the tertiary C–H bond and the secondary C–H bond α to the nitrogen.

At the same time, as suggested by another reviewer, the reaction with cyclohexane formed methylcyclohexane in only 0.8% yield (by GC-FID with an authentic standard). This result suggests that the functionalization of unactivated secondary C(sp^3)–H bonds does not occur to a significant extent under our reaction conditions.

Our investigations suggest that Ni(acac)₂ becomes fully deactivated before Ir-F becomes fully deactivated. Therefore, we postulate that the catalytic activity of Ni(acac)₂ ultimately limits the yields of the reaction. Although the addition of 2,4-pentanedione increases the yields to 40-60%, we agree that there is still room for improvement. The main factor determining these yields is the efficiency of the Ni catalyst in intercepting the tertiary radical and transferring the methyl group onto the tertiary radical. This in turn depends on the catalytic longevity of the Ni catalyst.

(3) The author compares the outcomes of some reactions with those reported under previously documented reaction conditions, such as, in the synthesis of compounds 2s, 2u, 2v, and 2ab, where the experimental exceed those reported under established reaction conditions. Could you offer a reasonable explanation for this discrepancy?

Authors' response:

Our work targeted the methylation of unactivated C–H bonds, whereas the prior work led to the methylation of activated C–H bonds that are benzylic or α to a heteroatom. Thus, it is not surprising that the two different sets of conditions have different outcomes on unactivated tertiary C–H bonds. The major difference is the identity of the nickel catalyst. Ni(acac)₂ is more effective for C(sp³)–C(sp³) couplings between two sterically distinct carbon radicals, especially between a tertiary radical and a primary radical. This observation is consistent with various reports (Ref 16, 43–48), and it can be attributed to a homolytic radical substitution (S_H2) mechanism, as suggested by the extensive studies reported in these references.

(4) Comparison through entries 1-5 in Figure 2b, the condition that a mixture of dicumyl peroxide and cumyl tert-butyl peroxide in a solvent mixture of TFE and MeCN should be added in the manuscript.

Authors' response:

The suggested experiment has been performed and the outcome has been added to Figure 2b (entry 7). These conditions gave rise to the methylated product in 46% yield, which is lower than our best condition (50% yield).

(5) Please carefully check the main text and Supplementary Information, there are a number of errors and inconsistencies:

a) The yield of compound 2x is inconsistent between the manuscript and ESI.

Authors' response:

The typo in the ESI has been fixed. The yield of **2x** on page 46 has been corrected from 52% to 59%; the mass remains the same (45.1 mg).

b) In the optimization of the reaction conditions, the yield of compound 2a is inconsistent between the manuscript (Table 1, entry 6) and ESI (Section 3.1, entry 7).

Authors' response:

The typo in the ESI has been fixed. The yield of Section 3.1, Entry 7 has been corrected from 17% to 18%, and the yield of Section 3.1, Entry 8 has also been corrected from 18% to 17%.

c) Condition b should be noted in Section 3.10 in the ESI.

Authors' response:

It is unclear to which condition the reviewer was referring. But for consistency purposes, the conditions in the manuscript (Figure 2b and Table 3) have all been added to Section 3.10 of the SI.

d) The HRMS analysis is highly biased for compound 4c.

Authors' response:

We appreciate the reviewer pointing out this typo. It has been corrected from 234.1290 to 234.1490.

e) Reference punctuation formatting symbols should be standardized.

Author's response:

The reference formatting has been standardized and double-checked for errors.

(6) The following compounds are not pure, and therefore need to be re-purified, such as, compounds 1q, 1ab, 4b, 2a, 2z.

Authors' response:

We have repurified the stated compounds and updated the spectra for 1q, 1ab, 2a, and 2z. For 2a and 2z, we have also updated the corresponding yields in both the manuscript and the SI.

Despite multiple attempts to purify substrate 4b, we were unable to obtain analytically pure samples. It is likely that 4b is unstable, and minor impurities are formed during the purification by column chromatography. These impurities are present in the regions of 1-1.5 ppm and 3.5-4.5 ppm of the ^1H NMR spectrum. However, these impurities amount to <5% of the product and did not appear to have a large effect on the yield of the C–H methylation reaction. The yield of 5b from 4b is comparable to that of 5a and 5c from 4a and 4c.

REVIEWERS' COMMENTS

Reviewer #1 (Remarks to the Author):

The authors have thoroughly addressed all reviewers' comments and concerns, significantly improving the manuscript. This high-quality work is now ready for publication. However, I suggest incorporating one additional relevant citation:

During the revision process, a pertinent study on Ni/Photoredox catalyzed α -C(sp³)-H (trideutero)methylation and alkylation was published (Nat. Catal. 2024, <https://doi.org/10.1038/s41929-024-01192-7>). The inclusion of this reference would provide relevant context and further support the significance of the current study.

Given the timely nature of this work, I recommend proceeding with publication as soon as possible without requiring additional review from me.

Reviewer #2 (Remarks to the Author):

The authors have addressed all my questions, I recommend accepting this paper.

Reviewer #3 (Remarks to the Author):

The authors have adequately addressed all the concerns I had. I am satisfied with the revised version and would recommend publishing it in this journal as it stands.